# LOVD: Large-and-Open Vocabulary Object Detection

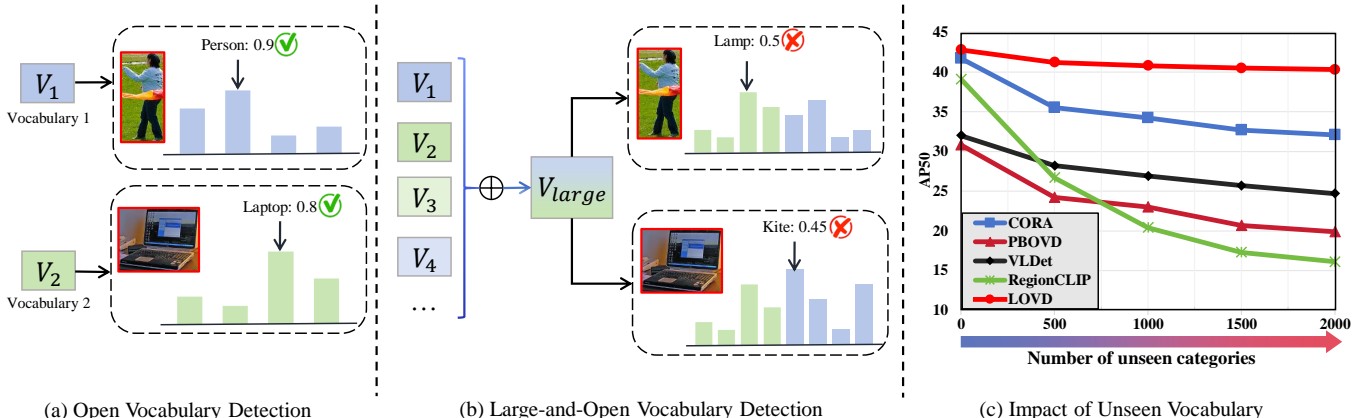

(a) Open Vocabulary Detection     (b) Large-and-Open Vocabulary Detection     (c) Impact of Unseen Vocabulary

**Figure 1: Motivation of our method. (a) Most existing open-vocabulary detectors rely on a precise and compact vocabulary for accurate prediction. (b) Their performance becomes less reliable as the input vocabulary expands. (c) The impact of the vocabulary size on detection performance. The proposed LOVD method outperforms existing open-vocabulary detectors on COCO dataset with large vocabularies.**

## ABSTRACT

Existing open-vocabulary object detectors require an accurate and compact vocabulary pre-defined during inference. Their performance is largely degraded in real scenarios where the underlying vocabulary may be indeterminate and often exponentially large. To have a more comprehensive understanding of this phenomenon, we propose a new setting called Large-and-Open Vocabulary object Detection, which simulates real scenarios by testing detectors with large vocabularies containing thousands of unseen categories. The vast unseen categories inevitably lead to an increase in category distractors, severely impeding the recognition process and leading to unsatisfactory detection results. To address this challenge, We propose a Large and Open Vocabulary Detector (LOVD) with two core components, termed the Image-to-Region Filtering (IRF) module and Cross-View Verification ($CV^2$) scheme. To relieve the category distractors of the given large vocabularies, IRF performs image-level recognition to build a compact vocabulary relevant to the image scene out of the large input vocabulary, followed by region-level classification upon the compact vocabulary. $CV^2$ further enhances the IRF by conducting image-to-region filtering in both global and local views and produces the final detection categories through a two-branch voting mechanism. Compared to the prior works, our LOVD is more scalable and robust to large input

*MM '24, 28 October - 1 November 2024, Melbourne, Australia*
© 2024 Copyright held by the owner/author(s). Publication rights licensed to ACM.
ACM ISBN 978-1-4503-XXXX-X/18/06
https://doi.org/XXXXXXX.XXXXXXX

vocabularies, and can be seamlessly integrated with predominant detection methods to improve their open-vocabulary performance. Source code will be made publicly available.

## CCS CONCEPTS

• **Computing methodologies → Object detection**.

## KEYWORDS

Large-and-Open Vocabulary, Object Detection

## 1 INTRODUCTION

Open-vocabulary object detection [49] advances conventional detectors with the capability to recognize and localize novel object categories that are unseen during training. Prevalent open-vocabulary detectors are often backend by Vision-Language (VL) models [33] pre-trained on large-scale unlabeled image-text pairs. They mostly consist of the category-agnostic detection followed by the open-vocabulary classification, where the latter amounts to a cross-modal matching between visual and textual features in the aligned feature space. Though promising progress has been achieved, existing methods entail a precise and compact input vocabulary as a prerequisite for inference, where the object categories in the vocabulary are deliberately selected based on their relevance to the test scenarios (See Figure 1(a)). However, such a setting may be infeasible in real-world applications especially when the prior information of the test scenarios is absent.

To bridge the above gap, one straightforward idea is constructing a scenario-independent vocabulary during inference by exponentially enlarging the vocabulary size to cover all potential object categories that commonly appear across a wide scenarios. However, empirical evaluations in most existing studies are conducted with

limited input vocabulary sizes (*e.g.*, 17 categories in COCO [28] and 337 categories in LIVS [13]). It is not clear whether state-of-the-art open-vocabulary detectors can still maintain their performance given exponentially enlarged vocabularies as input.

In this paper, we put the above issue under the lens by introducing a new object detection task called large-and-open vocabulary detection. To simulate real-world scenarios, we extend the input vocabulary from dozens or hundreds to around two thousands during inference, encompassing frequently encountered object categories, while the training setup remains consistent with the conventional open-vocabulary detection (See Figure 1(b)). Our experiments observe a substantial decline in detection accuracy of existing methods [10, 27, 45] as the size of the category vocabulary increases (See Figure 1(c) as an example). This outcome aligns with expectations as the detection accuracy primarily hinges on visual-textural feature matching, and more categories in input vocabularies will inevitably lead to an increase in distractors, thereby escalating the complexity of feature matching and classification. The above drawbacks are suffered by most existing methods, which severely limits their application scenarios.

To remedy the above drawbacks, we present the Large-and-Open Vocabulary Detector (LOVD), a new detection paradigm which is more scalable and robust to large input vocabularies, and can be seamlessly integrated with predominant detection methods to improve their open-vocabulary performance. One of our key contributions is the Image-to-Region Filtering (IRF) module. To alleviate the impact of category interference in large vocabularies, IRF performs image-level recognition first to build a more compact vocabulary out of the large one by filtering out a significant amount of distracting categories and identifying those highly relevant to the input scene. A more fine-grained region-level classification is then applied to each object proposal based on the compact vocabulary. By adopting the above coarse-to-fine philosophy, IRF can effectively tackle the challenge brought by large vocabularies and yields more accurate detection performance in an efficient manner. In addition, since the vision encoders of most VL models are pre-trained on the image level, directly transferring them to region-level classification may encounter significant generalization issues. In comparison, IRF combining both image and region-level recognition provides an alternative solution to ensure better generalization ability of pre-trained VL models.

Most existing open-vocabulary detectors perform visual-textual feature matching in a global manner, *i.e.*, each object region is represented by a global feature vector, which offers greater flexibility but may ignore detailed visual cues. In contrast, local approaches have also been explored recently in image recognition [50], which align textual features with individual local image patches. While local matching excels at capturing granular details, it might not fully grasp the holistic context of the image. To harness the strengths of both global and local approaches, we propose the Cross-View Verification ($CV^2$) scheme to further enhance our IRF module, which performs image-to-region filtering in both global and local views, fostering an elegant cross-view interaction. Their predicted results are selected through a multi-branch voting mechanism to produce the final detection categories.

In summary, the contribution of this paper is threefold.

- We introduce the concept of large-and-open vocabulary detection, a problem that holds significant relevance to real-world scenarios yet remains largely unexplored within the community.
- We propose LOVD, a new open-vocabulary detection method, which can perform image-to-region filtering with cross-view verification, yielding more accurate detection results, especially with large vocabularies.
- We conduct extensive experimental evaluations across multiple datasets, which have justified the effectiveness of our method.

With its superior performance and adaptability, LOVD is well-positioned to meet the demands of real-world detection tasks, paving the way for more intelligent and versatile vision understanding. Source code and pre-trained models will be released.

## 2 RELATED WORK

### 2.1 Open Vocabulary Detection

The open vocabulary detection approach, pioneered by OVR-CNN, uses captioning data to associate novel semantic categories with visual regions [49]. Capitalizing on the success of pre-trained VLMs, which merge extensive image and language vocabularies, subsequent research has harnessed these models for OVD enhancements [20, 24–26, 33]. The first to implement CLIP in this domain, ViLD introduced a method for instance-level visual-to-visual knowledge distillation [12]. Following this, the DETR-style OV-DETR model utilized VLMs to create adaptive queries [48]. Later models, including CORA [45], incorporated region prompting [46] and anchor pre-matching to expedite training and better align disparate image data scales. Further strategies [5, 10, 22, 27, 31, 32, 37, 40, 42, 44, 51–53] to address data imbalances and feature misalignments have involved the use of more balanced datasets, pseudo labels, and expansive region-text pre-training. SIC-CADS [8] exploits global knowledge derived from CLIP to substantially refine existing OVD models. Our innovation, LOVD, specifically targets the refinement of image-to-region filtering to effectively manage the complexities of extensive vocabularies.

### 2.2 Vision-Language Models

Vision-Language Models (VLMs) have seen transformative developments with the advent of models like CLIP [33], Align [20], and COCA [47], which harness contrastive learning to analyze expansive image-text datasets. These models are renowned for their ability to offer an in-depth, nuanced understanding of images within a comprehensive contextual framework, significantly surpassing traditional image analysis methods. VLMs play a critical role in diverse applications, from enhancing sophisticated image retrieval systems [1–3, 18, 21, 30] to developing innovative human-computer interaction techniques [34, 36, 39, 41, 43]. Innovations continue with models [4, 9, 11], which extend the capabilities of VLMs in synthesizing and interpreting text-image content. Additionally, these advancements facilitate the integration of VLMs into dynamic environments where adaptive learning and context-aware processing are crucial.

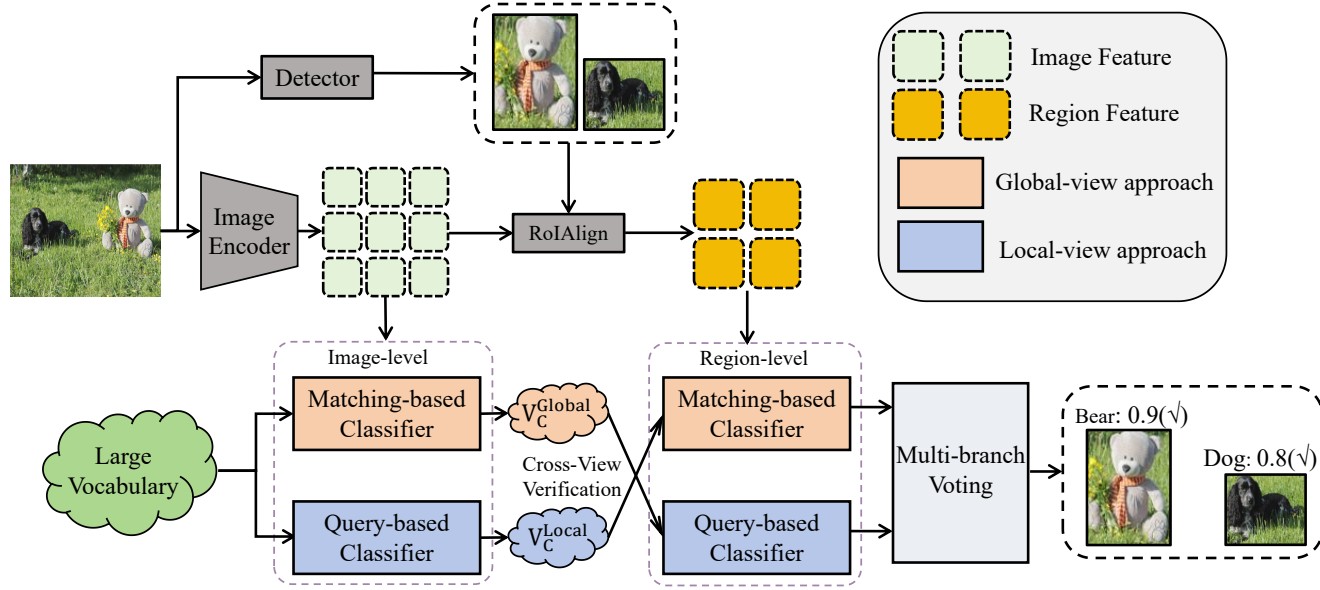

**Figure 2: Overview of our approach LOVD. It includes a category-agnostic localization module, an Image-to-Region Filtering (IRF) module, and a Multi-branch Voting module. The process begins with feature map extraction from an input image by a pre-trained visual encoder. Potential objects are localized by a region proposal network and their features are extracted via RoIAlign. The IRF module employs a Cross-View Verification ($CV^2$) scheme for object recognition against a comprehensive vocabulary, and the Multi-branch Voting module determines the final class label.**

## 2.3 Open Vocabulary Recognition

Open Vocabulary Recognition, also known as zero-shot multi-label recognition, is a task where the objective is to categorize multiple tags relevant to an image. Although vision-language models (VLMs) are adept at recognizing a broad range of open-set categories in single-label scenarios [6], their effectiveness diminishes in complex multi-label environments due to inadequate modality interactions. In contrast, models like RAM [50], Tag2Text [19], DualCoop [38], and others [15–17] have shown significant advancements in managing tasks with extensive vocabularies by leveraging large-scale image-tag data for training VLMs. These models exhibit robust zero-shot learning capacities, particularly excelling in detailed region-specific semantics rather than just focusing on global image attributes.

## 3 METHOD

### 3.1 Task definition

***Open-Vocabulary Detection.*** In open-vocabulary detection, a detector is trained using bounding box annotations and class labels of base categories $V_B$. During inference, given a pre-defined set of novel categories $V_N$, where $V_N \cap V_B = \emptyset$, the detector aims to detect all objects belonging to an open vocabulary $V_O = V_B \cup V_N$ containing both base and novel categories. To this end, existing open-vocabulary detectors often operate in a two-stage manner, *i.e.*, a category-agnostic localization step (*e.g.*, using Region Proposal Network [35]) followed by an open-vocabulary classifier. The text embeddings of category names extracted by a pre-trained VLM can

be employed as the classification weights, and the open-vocabulary classification is then equivalent to visual-textural feature matching. Compared to its closed-set counterparts, open-vocabulary detection is able to recognize unseen categories in a zero-shot manner, which significantly benefits a wider range of application.

***Large-and-Open Vocabulary Detection.*** Though open vocabulary detection has largely advanced the field, existing studies [8, 44, 45, 53] are mostly conducted under an ideal setting, where the input vocabulary during inference is precise and meticulously crafted according to the input scenarios. This may be significantly diverged from real applications, as the prior information of testing scenarios can be insufficient to construct a compact and curated vocabulary. To circumvent this issue, a simple workaround is to extend the input vocabulary to include as many potential categories as possible. Therefore, it is natural and important to investigate the task of large-and-open vocabulary detection. Specifically, its training setting is the same as conventional open-vocabulary detection. During inference, the input vocabulary becomes exponentially larger and uncrafted, containing thousands of commonly encountered object categories, most of which do exist in the input image. Section 4.2 presents the detailed procedure on how to build such a large inference vocabulary for existing detection datasets. Large-and-open vocabulary detection has the potential to bridge the gap between real-world requirements and conventional evaluation settings, and is able to more comprehensively investigate open vocabulary detectors.

## 3.2 Method Overview

In this section, we present the details of our proposed Large-and-Open Vocabulary Detector (LOVD) model. As shown in Figure 2, it mainly consists of a category-agnostic localization module, a Image-to-Region Filtering (IRF) module, and a two-branch voting module. Given an input image, we first extract the feature map using the visual encoder of a pre-trained VLM. All potential objects are localized using an off-the-shelf region proposal network [35], and their features are extracted with RoIAlign [14] from the image feature map. The IRF module then performs recognition for each object proposal based on the input vocabulary via a Cross-View Verification ($CV^2$) scheme. The final class label is determined by the two-branch voting module. In the following, we will elaborate on the implementation of each module.

## 3.3 Image-to-Region Filtering

As the input vocabulary becomes exponentially larger, the category interference issue arises as a unique challenge of large-and-open vocabulary detection. Since object recognition needs to consider significantly more candidate categories with most of them acting as noisy distractors, the open-vocabulary recognition results will become less reliable. To mitigate this issue, we borrow a coarse-to-fine philosophy and propose the Image-to-Region Filtering (IRF) module, which divides the open-vocabulary object recognition into two steps: image-level selection and region-level recognition. These two steps essentially involve two open-vocabulary classifiers with different purposes. Among them, image-level selection performs a one-pass recognition for the entire input image and aims to identify all existing object categories in the image. These potentially existing categories will be selected from the input large vocabulary ($V_L$) to constitute a pruned and more compact vocabulary ($V_C$) with a significantly reduced size. In the region-level recognition step, another classifier will take the new vocabulary as input to perform classification for each object proposals.

Although conceptually simple, the IRF module significantly benefits large-and-open vocabulary detection from the following two aspects. Firstly, previous study [45] shows that it is non-trivial to transfer an image-level pre-traiend VLM to region-level classification. The IRF module with an image-level selection step can naturally alleviate the above generalization gap and effectively prune the input large vocabularies, ensuring more accurate final classification results. Secondly, the enlarged vocabulary size will linearly improve computational burden. Taking the cosine similarity computation between $N$ categories and $K$ object proposals as an example, the original compute count is $N \times K$. Thanks to the two-step coarse-to-fine pipeline, the cosine similarity for all the $N$ categories are only computed against the input image for one time in IRF, and the overall compute count becomes $N + MK$, where $M \ll N$ denotes the size of the pruned vocabulary. In our experiments with $N = 2000$, $K = 1000$, and $M = 15$, IRF effectively reduces 99.15% of the overall computational complexity.

## 3.4 Cross-View Verification

Though with different purposes, the core components of the two steps in IRF module are both open-vocabulary classifiers. Therefore, how to design these classifiers are critical. As explained in Section 1, existing methods are restricted to either global or local representations, and may fail to capture detailed visual cues or lack a holistic understanding of the input scenes. We believe these two kinds of methods may largely cooperates with each other. To this end, we design two open-vocabulary classifiers, namely, a matching-based and a query-based method. The matching-based classifier represents the input visual content from a global view, while the query-based method focus on local visual cues.

A Cross-View Verification ($CV^2$) scheme is further developed, which integrates the above two classifiers into the IRF module in a cooperative manner to further enhance the open-vocabulary recognition performance for large input vocabularies. As shown in Figure 3, both the image-level selection and region-level recognition step contain a matching-based and a query-based classifier in parallel. In image-level selection, the two classifiers will produce two pruned vocabularies, which will then be respectively received by a different classifier in the region-level recognition step. More specifically, the vocabulary generated by the matching-based (query-based) classifier in the image-level selection step will serve as input to the query-based (matching-based) classifier in the region-level recognition step. Finally, the IRF module will produce two separate classification results, which have been crossly verified by the two classifiers from both global and local views in different orders. As a consequence, we are able to achieve a better synergy between these two kinds of classifiers, giving rise to more reliable classification results.

***Matching-based Classifier.*** Similar to prior methods [45], the matching-based classifier performs open-vocabulary recognition by matching the similarity of the visual input (*i.e.*, either the input image or each object proposal) and the category names in the input vocabulary. It employs the pre-trained CLIP model [33] to extract visual and textual features, where the visual input is projected into a global feature vector $f$ and the vocabulary is mapped into the textual feature set $\{t_c\}_{c=1}^{N}$, where $c$ denotes the index category names in the vocabulary. Since the extracted visual-textual features have been aligned in a joint space, the classification results for the current visual input can be directly determined according to its cosine similarity score to each category name as follows:

$$m_c = \frac{\exp\left(\cos\left(f, t_c\right)\right)}{\sum_{c' \in V_O} \exp\left(\cos\left(f, t_{c'}\right)\right)}. \tag{1}$$

***Query-based Classifier.*** To preserve fine-grained visual cues, the query-based classifier divides the visual input into $H \times W$ patches. A vision Transformer encoder is employed to convert the input patches into visual feature $F \in \mathbb{R}^{HW \times d}$, with $d$ indicating the feature dimension. Meanwhile, the category names in the vocabulary are also encoded into textual features $\{t_c\}_{c=1}^{N}$ using pre-trained VLM. A lightweight Transformer decoder is further adopted, which takes the textual features as queries and performs cross-attention between textual and visual features to generate a set of textual embeddings $\{e_c\}_{c=1}^{N}$. Finally, each textual embedding is passed through a linear layer followed by a Sigmoid activation to predict the probability score $q_c$ for the corresponding category.

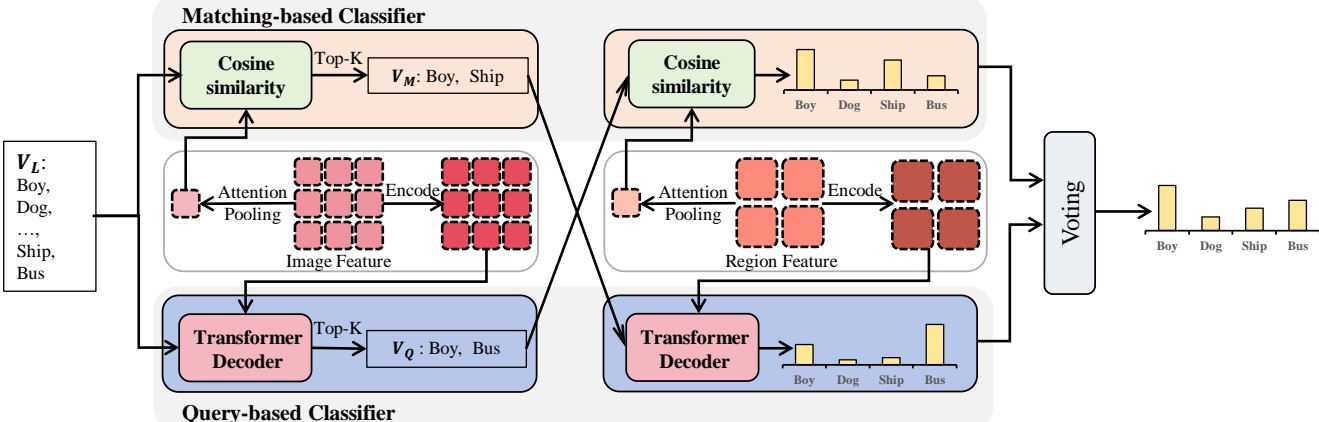

**Figure 3: Overview of the Cross-View Verification (CV$^2$) scheme. Two classifiers each produce a compact vocabulary ($V_C$), which is then processed by another classifier. The final classification result is computed through a voting process that incorporates these two distinct classification outcomes.**

### 3.5 Training and Inference

**Training.** We adopt an off-the-shelf category-agnostic detector for object localization. The visual and textual encoders are pre-trained through contrastive learning following [33]. The matching-based classifier is based on these two encoders and does not require any further fine-tuning on object detection data. The query-based classifier is trained on both image-level and region-level classification. For image-level training, we follow [50] to collect image-level pseudo labels. For region-level training, groundtruth bounding box annotations are directly used as training labels. Training is conducted by optimizing the following binary cross-entropy loss:

$$\mathcal{L}_{\text{cost}}(y, q) = -\frac{1}{N} \sum_{c \in V_B} \left[ y_c \cdot \log(q_c) + (1 - y_c) \cdot \log(1 - q_c) \right], \quad (2)$$

where $q_c$ denotes probability score of the $c$-th category predicted by the query-based classifier, and the groundtruth label $y_c = 1$ indicates that the $c$-th category exists in the input image and $y_c = 0$ otherwise. Since the IRF module with the CV$^2$ scheme is compatible with most predominant open-vocabulary detectors, it can be easily applied to and significantly benefits existing methods as shown in our experiments.

**Inference.** For each input image, the matching-based and query-based classifier in the image-level selection step will first produce the refined vocabulary $V_M$ and $V_Q$, which contain the top $K$ categories with the highest scores predicted by the two classifiers, respectively. In the region-level recognition step, for each object proposal, the two classifiers will separately predict a probability score for each of the categories in the refined vocabularies. To obtain the final classification results, we design a two-branch voting mechanism as follows.

$$p_c = \begin{cases} m_c \times q_c, & \text{if } c \in V_M \cap V_Q, \\ (m_c)^{\alpha}, & \text{if } c \in V_Q \text{ and } c \notin V_M, \\ (q_c)^{\beta}, & \text{if } c \in V_M \text{ and } c \notin V_Q, \\ 0, & \text{otherwise}, \end{cases} \quad (3)$$

where $m_c$ and $q_c$ are probability scores for the $c$-th category predicted by the matching-based and query-based classifiers, respectively. The hyperparameters $\alpha$ and $\beta$ are used to balance the confidence of the two classifiers. The final classification result is determined as the object category with the highest class score $p_c$.

## 4 EXPERIMENTS

In this section, we extensively evaluate the proposed LOVD method across various open-vocabulary and large-and-open vocabulary detection tasks. Details about the datasets and evaluation metrics are presented in Section 4.1, and the specific implementation nuances of our LOVD are outlined in Section 4.2. We perform comparisons with leading methods in Section 4.3, highlighting the superior capabilities of our approach. Further, we conduct the ablation study to investigate the IRF module, the CV$^2$ scheme, and hyperparameters in Sections 4.4.

### 4.1 Datasets & Evaluation Metrics

We conduct experiments on the two popular benchmark datasets, including COCO [28] dataset and LVIS [13] dataset. To verify the robustness of the compared detectors, both the open-vocabulary and large-and-open vocabulary settings are adopted for the two datasets. More detailed explanations are as follows.

**Open-vocabulary Setting.** Following the conventional setting [3], the vocabulary of COCO dataset is partitioned into 48 base categories for training, and 17 novel categories only for testing the open-vocabulary ability of models. Similarly, the LVIS dataset includes 1,203 categories, classified as frequent, common, and rare categories based on their occurrence. Following prior methods [12], the frequent and common categories are used for training, and the rest 337 rare categories are adopted as novel categories for testing.

**Large-and-Open Vocabulary Setting.** The only difference between the open-vocabulary setting and large-and-open vocabulary setting lies in the input vocabulary during inference. For both COCO and LVIS datasets, we augment the size of their original

| Method | Pre-train Model | Params(M) | LOV | | | OV | DR (%) |
| | | | Novel | Base | All | Novel | Novel |
|---|---|---|---|---|---|---|---|
| Region CLIP [52] | CLIP (RN50) | 160 | 15.1 | 43.4 | 36.0 | 31.4 | 44.3 |
| Region CLIP [52] | CLIP (RN50x4) | 285 | 16.7 | 38.2 | 32.6 | 39.3 | 42.5 |
| OV-DETR [48] | CLIP (RN50x4) | 421 | 20.9 | 48.3 | 28.1 | 29.4 | 71.1 |
| PB-OVD [10] | CLIP (ViT-B/32) | 189 | 19.9 | 41.3 | 35.7 | 30.8 | 64.6 |
| VLDet [27] | CLIP (RN50) | 141 | 24.7 | 46.5 | 40.8 | 32.0 | 77.2 |
| VLDet [27] + LOVD | CLIP (RN50) | 155 | 29.4(+4.7) | 49.0(+2.5) | 43.9(+3.1) | 34.8(+2.8) | 91.2(+14.0) |
| BARON [44] | CLIP (RN50) | 201 | 31.7 | 54.8 | 48.8 | 34.7 | 91.3 |
| BARON [44] + LOVD | CLIP (RN50) | 215 | 33.8(+2.1) | 56.2(+1.4) | 50.2(+1.4) | 36.2(+1.5) | 93.4(+2.1) |
| Detic [53] | CLIP (RN50) | 81 | 25.4 | 49.4 | 36.8 | 27.8 | 91.4 |
| Detic [53] + SIC-CADS | CLIP (RN50) | 157 | 29.3 (+3.9) | 51.4 (+2.0) | 39.7 (+2.9) | 31.0 (+3.2) | 94.5 (+3.1) |
| Detic [53] + LOVD | CLIP (RN50) | 95 | 32.3 (+6.9) | 49.6 (+0.2) | 39.8 (+3.0) | 32.8 (+5.0) | 98.5 (+7.1) |
| CORA [45] | CLIP (RN50) | 114 | 27.6 | 25.4 | 26.0 | 35.1 | 78.6 |
| CORA [45] + LOVD | CLIP (RN50) | 128 | 36.4 (+8.8) | 32.7 (+7.3) | 33.7 (+7.7) | 38.9 (+3.8) | 93.6 (+15.0) |
| CORA [45] | CLIP (RN50x4) | 190 | 32.1 | 31.7 | 31.8 | 41.7 | 77.0 |
| CORA [45] + LOVD | CLIP (RN50x4) | 204 | 40.3 (+8.2) | 37.9 (+6.2) | 38.5 (+6.7) | 43.3 (+1.6) | 93.1 (+16.1) |

Table 1: Main results on the COCO dataset. We report AP50 as the evaluation metric. "LOV" and "OV" denote the large-and-open vocabulary and open-vocabulary settings, respectively.

vocabulary up to a total of 2,000 categories. The newly introduced categories pose more challenges for object recognition compared to the open-vocabulary setting.

**Evaluation Metrics.** Following the prior methods [45, 53], we utilize the $AP50$ of bounding boxes as the evaluation metric for the COCO dataset. This metric calculates the average precision at an intersection over union (IoU) threshold of 50% for each category and then computes the overall average across all categories. In the open-vocabulary setting, we specifically compute the $AP50$ for the novel categories. In the open-and-large vocabulary setting, we assess the performance across novel, base, and all categories.

For the LVIS dataset, we compute the $mAP$ of masks averaged on IoUs from 0.5 to 0.95. Similar to the COCO dataset, we compute the $mAP$ for the novel categories for the open-vocabulary setting, while perform evaluation for rare (novel), common, and frequent categories in the open-and-large vocabulary setting.

In addition, We are also interested in comparing the performance degradation on the novel categories of detectors from open vocabulary to large-and-open vocabulary settings. Therefore, a relative metric named the Decay Ratio (DR) is computed as follows:

$$DR = \frac{\text{LOV-AP}_{\text{novel}}}{\text{OV-AP}_{\text{novel}}} \times 100\%, \qquad (4)$$

where LOV-AP$_{\text{novel}}$ and LOV-AP$_{\text{novel}}$ denote the AP value for novel categories under the open-vocabulary and large-and-open vocabulary settings, respectively.

**Cross-dataset Evaluation .** Following previous work [10, 27, 40, 49], we train our LOVD on the training set of COCO and conduct zero-shot transfer experiment on PASCAL VOC [7] to evaluate the generalization ability of the proposed method, using AP50 as the metric.

## 4.2 Implementation Details

**Building Large Vocabulary.** The number of object categories in both COCO and LVIS is limited for large-and-open vocabulary evaluation. To augment the existing vocabulary for these datasets, we apply two principled criteria: (1) The newly added categories must not overlap or be synonymous with any categories of vocabulary in the validation set. (2) They should be absent from all images in the validation set. Our selection starts by gathering more than 10,000 common categories from databases like OpenImages [23] and ImageNet-21K [6] as the initial list. Then, we utilize the multi-modal large language model [29] to eliminate categories from the initial list that already exist in the validation set, and employ CILP [33] to filter out the semantically redundant categories. Finally, we elaborately refine the categories manually to build the large vocabulary with 2,000 categories in total. For more details, refer to the supplementary materials.

**Training Setups.** We train the Query-based classifier on COCO with the supervision of pseudo tag labels generated by [50] with the binary cross entropy loss. To avoid label leakage, tags associated with the 17 novel categories are excluded from the pseudo-labels during training. Similarly, the training protocol on the LVIS dataset follows this principle that excludes categories related to rare categories. We train the query-based classifier on an NVIDIA Tesla A100 using PyTorch, employing the AdamW optimizer with an initial learning rate of $2 \times 10^{-5}$. The batch size is set to 4, and the training process runs for a total of 10 epochs. Based on the observations in our ablation studies (as shown in n Section 4.4), the value of hyperparameters $\alpha$ and $\beta$ in (3) are set to 1.0 and 4.0, respectively. The number of selected categories $K$ in image-level selection step is set to 15. We keep the above setting consistent for both COCO and LVIS datasets.

| Method | Pre-train Model | Params(M) | LOV | | | OV | DR (%) |
| | | | Novel | Common | Frequent | Novel | Novel |
| --- | --- | --- | --- | --- | --- | --- | --- |
| VLDET [27] | CLIP(RN50) | 184 | 20.3 | 29.1 | 33.6 | 21.7 | 93.5 |
| VLDET [27] + LOVD | CLIP(RN50) | 198 | 22.7(+2.4) | 30.0 (+0.9) | 33.4(-0.2) | 23.7(+2.0) | 95.8(+1.7) |
| Detic [53] | CLIP(RN50) | 81 | 23.8 | 32.0 | 35.3 | 24.9 | 95.5 |
| Detic [53] + SIC-CADS [8] | CLIP(RN50) | 157 | 25.1(+1.3) | 32.5(+0.5) | 35.3(+0.0) | 26.2(+1.3) | 95.8(+0.3) |
| Detic [53] + LOVD | CLIP(RN50) | 95 | 26.6(+2.8) | 32.6(+0.6) | 35.5(+0.2) | 27.3(+2.4) | 97.4(+1.9) |

Table 2: Main results on the LVIS dataset. We report mAP as the evaluation metric. "LOV" and "OV" denote the large-and-open vocabulary and open-vocabulary settings, respectively.

## 4.3 Overall Comparison

Since the main contribution of our LOVD is the IRF module with the $CV^2$ scheme, the other modules are compatible with most existing detectors. Therefore, to evaluate the effectiveness and adaptability of our LOVD, we integrate it with different prior open-vocabualry detectors that employ different network architectures and learning principles. These include prompt learning (CORA [45]), weakly supervision (Detic [53]), and region-text alignment (BARON [44], VLDet [27]). We present the comparative results of the original open-vocabulary models and their LOVD-enhanced counterparts (denoted with the postfix "+LOVD" in Table 1-3).

**Results on COCO**. Table 1 reports the quantitative results of the compared methods and the integration of LOVD with these models on the COCO benchmark dataset. All the state-of-the-art open-vocabulary models, including those trained with extensive vocabularies of 21K categories from ImageNet-21K [6] such as Detic [53] and Baron [44], show a significant decrease in performance under the large-and-open vocabulary setting. In contrast, integrated with the proposed LOVD, it consistently and significantly enhances the performance of all representative models in identifying novel categories across all the inference settings. Specifically, for the open-vocabulary setting, it shows that the integration of LOVD achieves an improvement of 5.0 in terms of $AP50$ for the Detic method [53]. For the large-and-open vocabulary setting, the performance improvement is more significant for the novel categories. It achieves a maximum improvement of 8.8 in terms of $AP50$ compared with the original methods, and establishes a new SOTA performance when incorporated into CORA. Moreover, LOVD exhibits great robustness towards large vocabulary in terms of decay ratio. With the assistance of LOVD, all representative methods achieve substantial improvements, with the smallest observed decline in performance being only 1.5%. These above results show that our proposed LOVD can effectively mitigate the category interference issue for the open-and-large vocabulary task.

**Results on LVIS**. LOVD demonstrates a similar superiority on the LVIS benchmarks, as detailed in Table 2. It shows that when expanding the categories of the input vocabulary, the existing leading methods VLDET [27] and Detic [53] exhibit a notable decline in performance under the large-and-open vocabulary setting. Nevertheless, the proposed LOVD consistently improves the performance of the two methods for all the evaluations. Specifically, it enhances

| Method | OV | LOV | DR (%) |
| --- | --- | --- | --- |
| PB-OVD [10] | 42.7 | 59.2 | 72.1 |
| VLDet [27] | 47.1 | 61.7 | 76.4 |
| CORA [45] | 51.9 | 65.9 | 78.8 |
| CORA [45] + LOVD | 54.3 (+2.4) | 66.6(+0.7) | 81.5(+2.7) |

Table 3: Zero-shot transfer results on the Pascal VOC. We report AP50 as the evaluation metric.

the *mAP* scores for the novel categories by more than 2.0 across these base models.

**Cross-dataset Generalization** . In the cross-dataset generalization experiment, we further conduct evaluation on the Pascal VOC dataset for the models trained on the COCO dataset. As shown in Table 3, integrated with our LOVD, it significantly improves the AP50 scores by 2.4 and 0.7 in both open-vocabulary and large-and-open vocabulary settings for the CORA method, and also improves the DR scores by 2.7. These results illustrate our remarkable enhancement in generalization ability across different datasets.

## 4.4 Ablation Study

We perform ablation studies to evaluate the key components of the proposed method. All the evaluations are conducted on the COCO dataset. "LOV-AP$_{novel}$" in Table 4-6 denotes the AP50 score for the novel categories under the large-and-open vocabulary setting.

**Image-to-Region Filtering**. We evaluate the performance of various IRF variants on top of the baseline method CORA [45] to showcase the significance of our IRF, as shown in Table 4. In the large vocabulary setting, the method proposed by SIC-CADS [8], which multiplies region-level and image-level scores, demonstrates inferior performance compared to the method of filtering before classification. Our IRF module, integrating image-level selection with region-level classification, achieved the highest performance. Table 4 also shows the results of varying values of $k$ in Top-k operation of image-level recognition. We observe that in the scenes of the COCO dataset, typically 10-20 object categories appear, so select $k$ from the range of 10-20. When using a smaller $k$, too many ground truth categories are filtered out. However, if a large $k$ is used, too many interfering categories are fed into the region classifier, resulting in a significant impact from the large vocabulary. Since

| Method | LOV-AP$_{novel}$ |
|---|---|
| Baseline | 27.6 |
| Baseline + Image-level Score Multiplication | 33.9 |
| Baseline + Image-level Selection | **35.0** |
| Baseline + Top-10 Selection | 32.5 |
| Baseline + Top-15 Selection | **35.0** |
| Baseline + Top-20 Selection | 34.3 |

**Table 4: Ablation studies on the IRF module.**

| Method | LOV-AP$_{novel}$ |
|---|---|
| Matching-based Classifier (region-level) | 27.6 |
| w/ Query-based Classifier (image-level) | 35.0 |
| Query-based Classifier (region-level) | 12.4 |
| w/ Matching-based Classifier (image-level) | 13.0 |
| Adding | 35.3 |
| Two-branch Voting | **36.4** |
| Matching-based Classifier | 10.2 |
| Query-based Classifier | 14.4 |
| Combination with Voting | 10.7 |

**Table 5: Results of different variants of $CV^2$.**

$k$ = 15 achieves a good trade-off between ground truth categories and interfering categories and obtains the best AP.

***Cross-view Verification*** . As shown in Table 5, Different variants of $CV^2$ can be employed in our LOVD approach. We evaluate the performance of $CV^2$ variants on top of the baseline module the region-level Matching-based Classifier to showcase the significance of our $CV^2$. By integrating a image-level Query-based Classifier, we establish the first branch, which notably increases AP from 27.6 to 35.0. Similarly, we evaluated a Region-level Query-based Classifier as the baseline, and by adding an image-level Matching-based Classifier, we formed the second branch, increasing AP from 12.4 to 13.0. However, these approaches result in diminished performance, indicating that the individual branches alone do not yield optimal results. Merely combining the results from both branches yields an AP of only 35.3, indicating that the branches alone do not provide optimal outcomes. It is the combination of Two-branch Voting that achieves superior performance, with an AP of 36.4, underscoring the importance of synergistic integration. Furthermore, we evaluate configurations that omit the cross-view combination, which leads to a marked decrease in performance. This decline is attributed to error accumulation among identical models, resulting in compounded inaccuracies. These findings suggest that simply amalgamating modules without the strategic integration provided by $CV^2$ does not effectively leverage their complementary capabilities and may even be counterproductive.

***Ablation of Hyperparameter***. Table 6 presents the impact of adjusting the values of $\alpha$ and $\beta$ in Equation 3. These parameters act as penalty coefficients for the scores of categories that one branch

| $\alpha$ | $\beta$ | LOV-AP$_{novel}$ |
|---|---|---|
| 0.5 | 2.0 | 29.9 |
| 1.0 | 2.0 | **31.8** |
| 2.0 | 2.0 | 28.4 |
| 1.0 | 2.0 | 31.8 |
| 1.0 | 4.0 | **36.4** |
| 1.0 | 8.0 | 36.2 |

**Table 6: Ablation studies on hyperparameter $\alpha$ and $\beta$ on COCO benchmark.**

considers present while another branch deems absent. The higher the coefficients, the greater the penalty applied to the scores. If the coefficients are set too high, the scores of the correct categories may become excessively low. Conversely, if the coefficients are too low, the scores for incorrect categories might be too high. Optimal values of $\alpha$ and $\beta$ effectively reduce the likelihood of false negatives and false positives. We adopt the best-performing hyperparameters $\alpha$=1.0 and $\beta$=4.0 from the COCO experiments. Although the ablation studies are conducted on the COCO dataset, given the similar scale of scores between COCO and LVIS datasets, we apply $\alpha$=1.2 and $\beta$=4.0 as the hyperparameters for the LVIS dataset.

***Scale of Vocabulary***. We investigate the effect of introducing a variable size of large vocabulary on model performance, as depicted in Figure 1(c) (See the supplementary materials for more details). When a modest number of 500 extra categories are introduced, OV models experience a notable decrease in accuracy. As we incrementally increase the count of extra categories from 0 to 2000, the performance of OVD methods consistently worsens, while our LOVD remains largely unaffected. This evidence indicates that our approach possesses a superior adaptability in facing real-world scenarios, demonstrating resilience against the influx of distracting categories.

## 5 CONCLUSION

In this paper, we introduce a novel task of large-and-open vocabulary detection, crucial for enhancing the real-world applicability of open-vocabulary object detection systems. To effectively manage the challenges posed by vast input vocabularies, we propose a new open-vocabulary detection method LOVD. By incorporating the Image-to-Region Filtering (IRF) module and the Cross-View Verification ($CV^2$) scheme, LOVD significantly improves detection accuracy and robustness against the interference of large vocabularies. Our extensive experiments on various datasets underscore LOVD's distinct advantage over existing approaches, showcasing its enhanced scalability and robustness in handling large vocabularies.

In the future, we will further expand the scale of the vocabulary to better simulate real scenarios under the large-and-open vocabulary detection setting. In addition, we will also experiment with more advanced VLM and network architectures to improve the performance and adaptation our of method.

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
