# OpenReview forum: "LOVD: Large Open Vocabulary  Object detection"
_acmmm.org/ACMMM/2024/Conference — MM2024 Poster_

### Official Review · Reviewer_WzNv · 2024-05-05

**Rating:** 5
**Confidence:** 4

**Summary:**

The number of unseen categories in the open-vocabulary object detection task is now normally limited. The performance of existing methods degrades when the number of unseen categories increases, especially when the number grows to extremely large. To tackle this problem, this paper formulates a new task named Large-and-Open Vocabulary Object Detection. Along with this new setting, a new detector named LOVD is proposed, which contains an Image-to-Region Filtering (IRF) module and the Cross-View Verification (CV2) scheme.

**Strengths:**

1.	The problem that this paper is trying to solve is important and necessary, which is the trend for future work.
2.	The newly proposed method LOVD is simple yet effective, achieving good performance compared with existing state-of-the-art methods.
3.	The parameter of LOVD is not very large, bringing little computational cost when inserting into existing methods.
4.	Authors claim that the source code will be made publicly available. Hoping the generated dataset on COCO and LVIS can also be public to serve as a good baseline for the research community.

**Limitations:**

1.	How to decide whether the number of unseen categories is large enough? What is the standard to distinguish between normal and large regarding open vocabulary object detection?
2.	Although the amount of parameters that LOVD brings is not very large, it is recommended to show the comparison of inference speed and training time to reflect the computational cost under real scenarios.
3.	The future work and limitations analysis are missing. This work, as the first paper for the new setting, is recommended to give more insight into future work and some analysis of the limitations of LOVD.
4.	The used template is not the latest one.

**Suitability:**

3

---

### Official Review · Reviewer_yQAq · 2024-05-13

**Rating:** 5
**Confidence:** 3

**Summary:**

This work aims to tackle a new problem, Large-and-Open Vocabulary Object Detection, in which thousands of unseen categories exist in the testing vocabulary. To address the challenge, this work proposes a coarse-to-fine method by first filtering some unrelated categories and building a compact vocabulary and then using the new image-specific vocabulary as the the classifier. Experiment results show improvements on both traditional open-vocabulary setting and Large-and-Open Vocabulary Object Detection setting.

**Strengths:**

1. This work tackles a new and challenging problem.
2. The proposed coarse-to-fine method and Cross-View Verification scheme can alleviate the classification difficulty.
3. The proposed method is general and can be applied to many existing open-vocabulary methods.

**Limitations:**

1. Figure1(c) illustrates the impact of the number of unseen categories for different open-vocabulary detectors during testing. However, it seems that the models are trained on OV-COCO. I wonder whether the same phenomenon can be found when the detector is trained on OV-LVIS with much more categories, especially detectors trained with federated loss. Since federated loss samples a small vocabulary during training while using all classes in the testing, which is similar to the Large-and-Open Vocabulary Object Detection setting.
2. In Table 4, the best k is 15. As the number of categories for some images in LVIS is possibly larger than 15, please provide the recall rate for each k, especially on LVIS dataset.
3. It seems that the proposed classifier needs many complex operations. Please compare the fps with the baseline.
4. The results in Table 5 is poor-organized and confusing. More experiment details should be provided.
5. In region-level recognition, different vocabularies are used in the matching-based and query-based classifier. I want to know the performance using the same vocabulary (the intersection or union of two vocabularies).

**Suitability:**

3

---

### Official Review · Reviewer_mbHP · 2024-05-21

**Rating:** 5
**Confidence:** 2

**Summary:**

The paper tackles open-vocabulary object detection (OVD) for large (more than 1k) vocabularies, in which standard solutions for OVD have degraded performance due to the high number of distractor categories. The authors propose a two-stage, two-branch module designed first to filter out most of the improbable classes using global image-level information and then reuse those modules on regions with the new reduced vocabulary.

**Strengths:**

- The tackled problem of large-scale open-vocabulary detection is interesting and relevant.
- The authors covered the relevant literature well.
- Experimental evaluation is extensive and shows improvements over SotA on OVD tasks both with and without enlarged vocabularies. The authors also performed ablation studies for most of the design choices.

**Limitations:**

- There are some omissions in the presentation and motivation of the task and the proposed solutions that could be improved, such as the contextualization of the task (what are the possible applications of LOVD? Do domain-specific applications often need smaller, specialized vocabularies, conversely?) and some design choices (why using two different open-vocabulary modules in IRF and in that configuration? How sensitive is the proposal to choosing the top categories K, \alpha, and \beta? How can we choose them for datasets other than the tested ones? )

- It seems the method is somewhat inspired by SIC-CADS [8] in the image-to-region filtering rationale, and thus I expected additional comparisons. Only Detic + SIC-CADS is considered, while in [8], results are available for other models in which also the proposed approach has been tested (e.g., BARON, CORA RN50x4)

- Have the authors recomputed SotA models to obtain numbers in Tables 1, 2, and 3, or have they reported numbers from other papers? I see small discrepancies (e.g., Detic + SI-CADS 26.2 (paper) vs. 26.5 [8]) that, if due to slight implementation differences, may hinder the significance of some small performance improvements reported.

Minor issues/typos:
- L99: backend --> backed
- L392: improve --> increase
- L399-400: what do the authors mean by overall computation complexity? Are they including backbone models in the 99.15% reduction?
- L405: are critical --> is critical
- L629: fix double LOV-AP
- L674: CILP --> CLIP

**Suitability:**

3

---

### Official Review · Reviewer_kM8i · 2024-05-22

**Rating:** 2
**Confidence:** 4

**Summary:**

This work introduces a new setting called Large-and-Open Vocabulary object Detection (LOVD), which tests object detectors with extensive vocabularies containing thousands of unseen categories. To address the challenges posed by this setting, the authors propose a Large and Open Vocabulary Detector (LOVD) featuring an Image-to-Region Filtering (IRF) module and a Cross-View Verification (CV2) scheme. These components work together to filter and classify relevant categories, making the detection process more scalable and robust in real-world scenarios.

**Strengths:**

1. The motivation for the paper is interesting and convincing
2. The completeness of the paper is very high, the writing is clear, and the experimental results are adequate

**Limitations:**

1. (Major) The choice of K in top-k selection seems to be sensitive to the final performance (Table 4). Especially, COCO and LVIS share the same image sources from COCO, the optimal selection of K introduced in this paper for them is 15 which may not fit the other datasets. In this case, how to handle the selection of K across different datasets, please verify.  (Minor) Additionally, is it possible to adaptively choose K?
2. (Minor) The method for constructing the Large-and-Open Vocabulary Setting may be inappropriate. The paper filters similar texts based on text feature similarity (supp. Lines 15-16) to ensure category diversity. However, using text similarity alone to determine semantic similarity and filter duplicate categories may be problematic. For example, using CLIP, the text feature cosine similarity between 'pool table' and 'dinner table' is 0.73, yet they are visually very different. (Major) However, more importantly, LOV appears to be constructed separately for different datasets (Lines 578-580). Therefore, the results obtained using different LOVs for different datasets may not be reliable. Could you provide a unified LOV set for various datasets (at least for COCO and LVIS)?
3. (Major) The proposed Image-to-Region Filtering method reduces the vocabulary from VL to VC by determining the potential categories present in the image (Lines 378-380). There are two concerns: 1) Fairness of comparison. The construction of LOV has already excluded highly similar texts, so Image-level selection can easily exclude newly added extra vocabulary (VL -> VC), making the proposed method's setting on LOV close to or even weaker than the OV setting ("typically 10-20 object categories appear", Lines 805-808) whereas other models handle the vocabulary size of VL, making it difficult to evaluate the superiority of the proposed methods. 2) The performance improvement for image-level selection shown in Table 4 is significant, likely due to the vocabulary filtering making the model easier to recognize. Please provide more analysis for validation.
4. (Minor) Inconsistent naming, VLDET in Table 2 and VLDet in Table 1.

**Suitability:**

2

---

### Meta-Review · Area_Chair_h2pf · 2024-07-02

**Recommendation:** Accept (Poster)
**Confidence:** 4

**Metareview:**

There is a consensus to accept this paper. The author's responses to the reviewers' comments didn't completely satisfy the reviewers.
In particular, there is still concern about both the effectiveness of the proposed method and its efficiency (FPS).
I suggest accepting this paper, given the general consensus.
However, I ask the authors to add some of the valuable Q&A to the paper or the supplementary for making new readers apart from reviewers available with these details.